Contrasting gene flow at different spatial scales revealed by genotyping-by-sequencing in Isocladus armatus, a massively colour polymorphic New Zealand marine isopod

Wells Sarah J. S.J.Wells@massey.ac.nz
Dale James
Evolutionary Ecology Group, Institute of Natural and Mathematical Sciences, Massey University , Albany , Auckland , New Zealand
Prentis Peter
Electronic publication date: 2018 Aug 22
Publication date: 2018
Volume: 6
Electronic Location ID: e5462
Received 2018 Mar 8; Accepted 2018 Jun 29
Copyright: ©2018 Wells and Dale
Copyright year: 2018
Copyright holder: Wells and Dale
License: This is an open access article distributed under the terms of the Creative Commons Attribution License, which permits unrestricted use, distribution, reproduction and adaptation in any medium and for any purpose provided that it is properly attributed. For attribution, the original author(s), title, publication source (PeerJ) and either DOI or URL of the article must be cited.
License URL: https://creativecommons.org/licenses/by/4.0/

Keywords: Gene flow, Population genomics, Genotyping-by-sequencing, Marine, Isopod, Direct development, Colour polymorphism

Funding: Massey University Research Fund This work was funded through a Massey University Research Fund to James Dale and Sarah Wells. The funders had no role in study design, data collection and analysis, decision to publish, or preparation of the manuscript.

==============================
Understanding how genetic diversity is maintained within populations is central to evolutionary biology. Research on colour polymorphism (CP), which typically has a genetic basis, can shed light on this issue. However, because gene flow can homogenise genetic variation, understanding population connectivity is critical in examining the maintenance of polymorphisms. In this study we assess the utility of genotyping-by-sequencing to resolve gene flow, and provide a preliminary investigation into the genetic basis of CP in Isocladus armatus, an endemic New Zealand marine isopod. Analysis of the genetic variation in 4,000 single nucleotide polymorphisms (SNPs) within and among populations and colour morphs revealed large differences in gene flow across two spatial scales. Marine isopods, which lack a pelagic larval phase, are typically assumed to exhibit greater population structuring than marine invertebrates possessing a biphasic life cycle. However, we found high gene flow rates and no genetic subdivision between two North Island populations situated 8 km apart. This suggests that I. armatus is capable of substantial dispersal along coastlines. In contrast, we identified a strong genetic disjunction between North and South Island populations. This result is similar to those reported in other New Zealand marine species, and is congruent with the presence of a geophysical barrier to dispersal down the east coast of New Zealand. We also found some support for a genetic basis to colouration evidenced by positive FST outlier tests, with two SNPs in particular showing strong association to the expression of a striped morph. Our study provides one of the first population genomic studies of a marine organism in New Zealand, and suggests that genotyping-by-sequencing can be a good alternative to more traditional investigations based on traditional markers such as microsatellites. Our study provides a foundation for further development of a highly tractable system for research on the evolutionary maintenance of CP.

Introduction

A key challenge in evolutionary biology is to understand the mechanisms maintaining genetic diversity in natural populations (Barton & Turelli, 1989; Brooks, 2002; Endler, 1986; Futuyma, 2005; Hartl & Clark, 1997; Hedrick, 1986; Hedrick, 2006; Kotiaho, Simmons & Tomkins, 2001; Lewontin, 1974; Nei, 1975; Pamilo, 1988; Roulin, 2004). Maintenance of diversity is paradoxical because both directional selection and genetic drift should act to remove most variation from populations (Ford, 1964; Futuyma, 2005; Hartl & Clark, 1997). However, diversity can be maintained in the presence of the homogenizing effect of gene flow among populations (Kawecki & Ebert, 2004). Therefore, resolving population structure in high gene flow species is critical to understanding how genetic variation is maintained within populations in the face of local adaptation. Research on colour polymorphism (CP), including how gene flow affects CP within and among populations, provides an excellent opportunity to gain insight into the evolutionary mechanisms that maintain biological diversity because CP is quantifiable and can occur on an immense scale (Roulin, 2004).

The spatial scales over which organisms and genes interact in the terrestrial and marine environments are often highly divergent. For example, the lack of physical barriers to gene flow, combined with the biphasic life cycles of many marine organisms, mean that the scales over which genetic structure may occur in the oceans can be considerably larger than in the terrestrial environment (Palumbi, 2004; Cowen & Sponaugle, 2009). These premises present challenges for effectively assessing gene flow in the marine environment, and are compounded by the large spatial scale, heterogeneous dynamic processes, and asymmetric flows of seascapes (Riginos et al., 2016).

Nevertheless, population genetic differentiation is generally expected to be greater in species with direct development than in organisms exhibiting biphasic life cycles, because of the reduced dispersal abilities of juveniles in these species (Benzie, 2000; Ross et al., 2009; but see Weersing & Toonen, 2009). The lack of a pelagic larval stage in the Isopoda, accompanied by the brooding of young and internal fertilisation, are purported to be limiting factors in their dispersal. Accordingly, population genetic studies in the Isopoda have revealed substantial population sub-structuring (Baratti, Filippelli & Messana, 2011; Baratti, Goti & Messana, 2005; Carvalho & Piertney, 1997; Hurtado, Mateos & Santamaria, 2010; Lessios & Weinberg, 1994; Markow & Pfeiler, 2010; Raupach et al., 2007; Raupach & Wägele, 2006; Sponer & Lessios, 2009; Xavier et al., 2011), as well as microgeographic variation in some intertidal species with specialist habitat requirements (Carvalho, 1989; Jolly, Rogers & Sheader, 2003; Piertney & Carvalho, 1994). However, population genetics of the New Zealand marine Isopoda has been overlooked and little is known of isopod dispersal dynamics around New Zealand.

Here, we provide an exploratory investigation of the genomic variation present within and between populations and colour morphs of a highly polychromatic New Zealand marine isopod, Isocladus armatus (Sphaeromatidae). The endemic I. armatus is abundant in intertidal rock pools on semi-sheltered shores throughout New Zealand. I. armatus presents an interesting case study for dispersal because, unlike other direct developing marine invertebrates studied in New Zealand (Knox, Hogg & Pilditch, 2011; Sponer & Roy, 2002; Stevens & Hogg, 2004), they are neither restricted by fragmentary estuarine habitats, nor by sedentary life histories. They are strong swimmers that are able to navigate their position in a water column (Morton & Miller, 1968). However, it is currently unclear whether this mobility enhances or reduces population genetic stratification. For example, this swimming ability could either enable them to resist displacement during turbulent tidal movements as suggested by Jansen (1968), or alternatively, to disperse substantial distances along the coast. Regardless, their widespread distribution throughout New Zealand and the Chatham Islands suggests that their dispersal ability can be substantial.

I. armatus displays extreme colour polymorphism (CP) within populations (Fig. 1). Whilst colouration varies across an almost continuous scale, certain colour morphs, observable throughout the species’ geographic range (Jansen, 1968), can be visually identified within this continuum (Fig. 1). Although the evolutionary drivers of CP in I. armatus are unknown, CP is generally presumed to have a genetic basis (Wellenreuther, Svensson & Hansson, 2014). Moreover, previous studies of the genetic association of pigmentation in Sphaeromatid isopods indicate that CPs are controlled by dominant alleles expressed at autosomal loci that are at low frequencies in the population (Hedgecock, Tracy & Nelson, 1982; Shuster, 1989). Although the extreme CP in I. armatus indicates a probable polygenic basis to colouration, it is likely that at least some of the more simple polymorphisms (such as presence/absence of a white dorsal stripe (Figs. 1F, 1G) are controlled by one, or a few, loci (e.g., Sassaman & Garthwaite, 1980). The extreme CP in I. armatus suggests its potential as a tractable system for understanding how genetic variation is maintained within populations. However, prior knowledge of the genetic basis of colouration, as well as population connectivity in this system, is crucial.

Figure 1 Colour polymorphism in I. armatus.

Within an almost continuous range of variation, certain colour morphs can be identified e.g., (A) and (B) variegated, (C) green, (D) white, (E) no pattern, (F) and (G) striped, and (H) spotted. Mature males (C), (E), and (H) possess a dorsal spine on the seventh pereonite. Photos by SJ Wells

Our study has two main objectives. First, we examine population structure and gene flow in I. armatus at two different spatial scales to validate the use of genotyping-by-sequencing (GBS) (Elshire et al., 2011) in investigating fine-scale and broad-scale dispersal in non-model marine species. To date, most studies of marine organisms have used microsatellites, or mitochondrial or nuclear DNA sequences to infer gene flow. However, single nucleotide polymorphisms (SNPs), abundantly and broadly distributed throughout the genome, can now provide a more thorough, genome-wide insight into genetic variation. In particular, reduced representation techniques such as GBS make it possible to genotype thousands of SNPs at relatively low costs in non-model species for which there is no reference genome available (Davey et al., 2011). Because small-scale genetic variation has been reported in other intertidal isopods, we predicted that we would see a low, but significant, level of population structure between two North Island populations 8 km apart. In contrast, we predicted strong differentiation between populations 1,000 km apart. Second, because information on the underlying genetic basis of colouration is critical to understanding how CP is maintained in this system, we conduct FST outlier tests of selection on this genomic data to determine if any loci present in our dataset are putatively associated with colour. This validation will form a basis for future studies to further resolve gene flow across multiple spatial scales, in combination with a detailed investigation of the naturally occurring phenotypic variability in this system.

Methods

Sample collection

To investigate population connectivity at two different spatial scales, we collected 31–32 isopods from two North Island populations 8 km apart (Hatfield’s Beach and Stanmore Bay Beach north of Auckland), and one South Island population (Kaikoura) around 1,000 km away (Fig. 2). From each location we aimed to collect an equal number of males and females. We collected approximately equal numbers of individuals of the four most easily classifiable morphotypes from each population: variegated, striped, (Jansen, 1968), white, and no-pattern (Fig. 1). No-pattern individuals were mostly plain brown or orange. However, within all morphotypes there was some additional variation in mottling or colour that was not assessed. Thus our morphs only capture the most apparent variations present in this species. All individuals whose morphotype could not be unambiguously assigned were excluded from the colour analyses (N = 12). Final numbers of each morph and population are listed in Table 1. Collected isopods were put into ice cold 70% ethanol and stored at −30 °C until extraction. All sample collection has been approved by a Ministry of Primary Industries New Zealand Special Permit 632.

Figure 2 Map showing sampling locations and the major ocean current systems around the New Zealand coast. Adapted from Heath (1985).

WAC, West Auckland Current; EAC, East Auckland Current; ECE, East Cape Eddy; ECC, East Cape Current; WE, Wairarapa Eddy; SC, Southland Current; TC, Tasman Current; WC, Westland Current; DC, Durville Current.

Table 1 Sample sizes of I. armatus populations and numbers of each colour morph sampled at each location.

	No pattern	White	Variegated	Striped	Unassigned	Total	
Hatfield’s Beach	10	5	7	3	6	31	
Stanmore Bay	8	7	8	8	1	32	
Kaikoura	12	3	8	3	5	31	

DNA extraction

Total genomic DNA was isolated from the head and pereopods of each sample. All DNA extractions were preceded by a step to deactivate nucleases modified from a protocol for bone samples in Edson et al. (2004). In this step, the head and pereopods of an individual were homogenised using a plastic pestle in a 1.5 mL Eppendorf tube in a 65 °C pre-heated solution consisting of 15 mL of 0.5M EDTA pH 7.5 per gram of sample to be digested, and 125 µL of 20% SDS per 1 mL of EDTA used. This was incubated and gently agitated overnight in an Eppendorf Thermomixer C at 65 °C. Following this step, total genomic DNA was isolated using a modified QIAgen DNeasy Blood and Tissue extraction protocol. The total volume of ATL buffer was increased to a ratio of no less than 2 parts ATL buffer to 1 part EDTA solution, and left to incubate overnight on a thermomixer at 65 °C instead of 56 °C. Two volumes of 12 µL Proteinase K were added to the lysis solution at least two hours apart. Total yields of genomic DNA were quantified using a Quantifluor ST fluorometer.

SNP genotyping

DArTseq™ represents a combination of a DArT complexity reduction methods and next-generation sequencing platforms (Courtois et al., 2013; Cruz, Kilian & Dierig, 2013; Kilian et al., 2012; Raman et al., 2014). Therefore, DArTseq™ represents a new implementation of sequencing of complexity reduced representations (Altshuler et al., 2000) and more recent applications of this concept on the next generation sequencing platforms (Baird et al., 2008; Elshire et al., 2011). Similarly to DArT methods based on array hybridisations, the technology is optimized for each organism and application by selecting the most appropriate complexity reduction method (both the size of the representation and the fraction of a genome selected for assays). The PstI-SphI complexity reduction method was selected for our data. DNA samples were processed in digestion/ligation reactions principally as per Kilian et al. (2012) but replacing a single PstI-compatible adaptor with two different adaptors corresponding to two different Restriction Enzyme (RE) overhangs. The PstI-compatible adapter was designed to include Illumina flow cell attachment sequence, sequencing primer sequence and “staggered”, varying length barcode region, similar to the sequence reported by Elshire et al. (2011). The reverse adapter contained flow cell attachment region and SphI-compatible overhang sequence. Only “mixed fragments” (PstI-SphI) were effectively amplified in 30 rounds of PCR using the following reaction conditions: 94 °C for 1 min followed by 30 cycles of 94 °C for 20 s, 58 °C for 30 s, and 72 °C for 45 s, with a final extension of 72 °C for seven minutes. After PCR, equimolar amounts of amplification products from each sample of the 96-well microtiter plate were bulked and applied to cBot (Illumina) bridge PCR followed by sequencing on Illumina Hiseq2500. The sequencing (single read) was run for 77 cycles resulting in fragments 77 bp long. Sequences generated from each lane were converted to FASTQ files using the Illumina HiSeq2500 software, and individuals were de-multiplexed based on the ligated barcodes. Each read was assessed using PHRED ( Ewing & Green, 1998) quality scores (Q-score), and any read containing Q-scores of <25 was removed. FASTQ files were then processed using proprietary DArT analytical pipelines. In the primary pipeline the FASTQ files were first processed to filter out poor quality sequences, applying more stringent selection criteria to the barcode region compared to the rest of the sequence. In that way the assignments of the sequences to specific samples carried in the “barcode split” step were very reliable. All reads were checked against the DArT database, as well as against GenBank bacterial and viral sequences to identify potential contaminations. Finally, identical sequences are collapsed into “fastqcall files”. These files are used in the secondary pipeline for DArT PL’s proprietary SNP and SilicoDArT (presence/absence of restriction fragments in representation) calling algorithms (DArTsoft14). The pipeline workflow is technically similar to the commonly used STACKS pipeline (Catchen et al., 2013). However, it differs from STACKS in that the sequence clusters are called for all pooled samples before being called for each individual. All monomorphic sequence clusters were removed, and SNPs were only called if both homozygous and heterozygous genotypes were identified. Multiple samples were processed from DNA to allelic calls as technical replicates and scoring consistency was used as the main selection criteria for high quality/low error rate markers. Any loci with very high read depths were also removed, retaining only SNPs with a high balance of read counts in allelic pairs, with a reproducibility of >95% and a minimum read depth of 3.

SNP filtering

The DArTseq GBS data resulted in a dataset of 26,215 loci. All putative SNPs possessed only two alleles. Further quality control filtering (Table S1) was applied to these loci in VCFtools v.0.1.13 (Danecek et al., 2011) to prepare the data for population genetic analyses. Only SNPs with a call rate (proportion of “missingness” across samples) ≥0.8 across all individuals were selected for further analysis. We retained loci with mean read depths of ≥5X across individuals, in order to eliminate loci where both alleles could not reliably be called. It has been demonstrated that low frequency alleles can create biases in population genetic analyses and tests of selection (Roesti, Salzburger & Berner, 2012). Therefore, we removed SNPs with a minor allele frequency (MAF) <0.05 both within a population, and SNPs with an average MAF <0.05 across the whole dataset ( Chakraborty, Fuerst & Nei, 1980). We also removed loci with observed heterozygosities >0.5 which could represent potential homeologs (Hohenlohe et al., 2011). Finally, putatively linked SNPs were removed by retaining only the SNP with greatest frequency of heterozygotes at each locus. For those analyses (STRUCTURE, MIGRATE) which assume Hardy–Weinberg equilibrium (HWE), we also removed loci displaying significant deviations (p < 0.01) from HWE in one or more populations.

Prior to all population analyses, loci potentially under either balancing or divergent selection were also removed (Table S1) as these “outlier” loci can produce different genetic signatures to neutral loci and bias estimates of population connectivity (Beaumont & Nichols, 1996; Luikart et al., 2003). However, these loci were retained in analyses aiming to detect loci under selection for colouration. Locus-specific departures from neutrality were assessed using two outlier FST methods. First, we employed the FDIST approach in ARLEQUIN. This method uses simulations to generate a null distribution of F-statistics, in which P-values are conditioned on observed levels of heterozygosities across loci (Excoffier, Hofer & Foll, 2009). We used the hierarchical island model (Excoffier, Hofer & Foll, 2009) with the North Island populations in group one and the Kaikoura population in group two. The hierarchical nature of this analysis takes into consideration asymmetric migration rates between populations due to spatial structuring or different population histories (Excoffier, Hofer & Foll, 2009). The analysis was carried out assuming 10 groups of 100 demes with 50,000 simulated loci. Loci were deemed under selection if they displayed a significant q-value after controlling for the False Discovery Rate (FDR) according to Storey & Tibshirani (2003). The FDR-adjusted q-values (FDR 5%) were calculated using the R package qvalue. The q-value of a locus defines the expected proportion of false positives among all loci with P-values equal to or less than the observed locus (Storey & Tibshirani, 2003). Second, we implemented a Bayesian approach in the program BayeScan (Foll & Gaggiotti, 2008) which uses population-specific FST coefficients (Beaumont & Balding, 2004) to calculate the posterior probability of a model including selection for each locus. The Bayesian approach employed in BayeScan has been shown to provide more reliable estimates of selection (Narum & Hess, 2011) due to the ability of posterior probabilities to correct for the identification of false positives to a greater extent than Bonferroni corrections (Foll & Gaggiotti, 2008). BayeScan was run using default parameters. From both analyses, loci displaying a q-value ≤0.05 were conservatively excluded from further population genetic analyses (Table S1).

Genetic structure and gene flow between populations

Global and per-locus observed heterozygosities, expected heterozygosities, and the population coefficient of inbreeding (FIS) were estimated for each population in the R package diveRsity using the function basicStats (Keenan et al., 2013). We used COANCESTRY (Wang, 2011) to estimate pairwise relatedness between North Island individuals in order to examine whether there were any differences in relatedness within versus between North Island populations. COANCESTRY was employed because it calculates relatedness based on seven different estimators and allows the user to determine the most appropriate estimator to use for the data. Because different marker-based relatedness estimators may perform better on different sets of loci due to variation between population composition and allele frequencies (Casteele, Galbusera & Matthysen, 2001), the most appropriate estimator to use for the data can be assessed in COANCESTRY by conducting simulations based on the real data. The most suitable estimator is that which produces relatedness values from simulations with the highest correlation coefficient to the actual data (Wang, 2011). TrioML, a maximum likelihood estimator was deemed the most suitable estimator for our data (r = 0.998) and was used to estimate differences in relatedness between groups. TrioML uses the genotype of a third individual as a reference in estimating the pairwise genetic relatedness between dyads, reducing the chance that genes identical in state are mistaken for genes that are identical by descent. We also used COANCESTRY to assess whether the mean relatedness within the two North Island populations is significantly different to the mean relatedness between the two North Island populations. Significance was assessed by bootstrapping over individuals from both populations using 10,000 resamples. The difference in mean group relatedness was deemed statistically significant if it fell outside the 2.5 and 97.5 percentiles generated by the permuted distribution (α = 0.05).

To quantify levels of genetic differentiation between populations, we calculated an unbiased estimator of pairwise FST (Weir & Cockerham, 1984) using the function pp.fst in the R package hierfstat (Goudet, 2005). The significance of these observed values was assessed by bootstrapping over 10,000 replicates and calculating 95% confidence intervals of these FST estimates using the function boot.ppfst.

We conducted Principal Coordinates Analysis (PCA) in the R package SNPRelate (Zheng et al., 2012) to visualise any genetic structure present in the dataset. PCA constructs low dimensional data projections with the aim of maximising the variance–covariance structure present among sampled genotypes. We conducted PCA on both the North Island dataset, as well as on all populations combined. A locus-by-locus hierarchical analysis of molecular variance (AMOVA) was performed in ARLEQUIN v.3.5.2.2 (Excoffier, Smouse & Quattro, 1992; Michalakis & Excoffier, 1996) with the two North Island populations grouped together to partition the total genetic variance among groups (North Island and South Island), among populations within groups (Hatfields Beach and Stanmore Bay), and within populations. Hierarchical AMOVA computations were also conducted using the amova function in the R package pegas (Paradis, 2010). This function requires as input a dist object representing a pairwise genetic distance matrix. We used the bitwise.dist function in the poppr package (Kamvar, Brooks & Grünwald, 2015; Kamvar, Tabima & Grünwald, 2014) to produce a dissimilarity matrix based on the fraction of different sites between sample genotypes according to Prevosti’s distance (Prevosti, Ocana & Alonso, 1975).

The number of genetic clusters (K) was inferred from individual assignments under a Bayesian framework implemented in STRUCTURE v.2.3.4 (Pritchard, Stephens & Donnelly, 2000). We ran the program with all populations, and then with the North Island populations only to confirm that any potential hierarchical signatures of population differentiation were not obscuring any weaker structuring that could be present in the North Island populations (Puechmaille, 2016). We ran the North Island analysis with information on sample location (LOCPRIOR model) included as a prior (Hubisz et al., 2009). These priors can be useful when location information is informative about true population clustering. This model uses information on the sampling locations to help tease out weak genetic structuring that STRUCTURE’s conservative algorithms might otherwise overlook, without being predisposed to identifying structure where there is actually none present. All models were run using a genetic admixture model and assuming correlated allele frequencies between populations. We specified K = 1 to K = 5 to be tested for the analysis with all three populations, and K = 1 to K = 4 to be tested for the analysis of two North Island populations only. Markov Chain Monte Carlo simulations were run for 100,000 iterations with a 50,000 burnin and 10 independent replicates. Assignment of individuals to clusters was inferred based on the inspection of a bar plot arising from the consolidation of results over the 10 replicates of K produced in CLUMPP (Jakobsson & Rosenberg, 2007) and visualised using DISTRUCT (Rosenberg, 2004). Optimal values of K were selected following the method of Evanno, Regnaut & Goudet (2005) in STRUCTURE HARVESTER (Earl & vonHoldt, 2012) which uses the change in log likelihoods between STRUCTURE results produced at different simulated K to infer the true K.

We used a randomly generated subset of 1,000 loci to investigate migration (M = m∕µ where m = immigration rate per generation, assuming a generation time of one year, and µ= mutation rate per site) between populations using coalescent modelling in MIGRATE (Beerli & Felsenstein, 2001). Due to the potentially large orders of magnitude difference in gene flow between the North Island populations and between the North and South Island populations, two analyses of migration were performed: the first investigated migration between Hatfield’s Beach and Stanmore Bay, and the second between the North Island populations (Hatfield’s Beach and Stanmore Bay combined) and South Island. For each analysis, we used a Bayesian Markov Chain Monte Carlo approach to investigate the probabilities of three different nested models based on alternative migration scenarios: (1) unidirectional gene flow from population 1 into population 2; (2) unidirectional gene flow from population 2 into population 1; and (3) bidirectional gene flow between the two populations allowing for asymmetric migration rates. Initial runs starting with mutation-scaled prior parameters for population size (θ) and the migration rate (M) based on FST values produced good posterior distributions, thus the mean values of 0.1 and 2500 on the priors for θ and M respectively were used in further analyses. We employed Metropolis sampling with a static heating scheme with four chains set at the default values, and ran each model for 1 million generations with 20 replicate chains sampling every 100 steps (to produce 20 million generations) with a burnin of 200,000 steps. Bayes factors were used to select the most appropriate model for the data (Beerli, 2006; Beerli & Palczewski, 2010).

FST outlier tests

Outlier FST tests were used to identify putative loci under selection and to investigate if any SNP loci are associated with a gene coding for colour. We conducted non-hierarchical tests of selection using morphotype as the grouping factor in Arlequin and BayeScan. Analyses were conducted for all populations together and for the North Island populations only, with the settings for the analyses the same as detailed above for the outlier analyses by population. We considered loci under potential selection to be those with a q-value of 0.05 or less (Storey & Tibshirani, 2003).

Results

Genetic structure and gene flow between populations

Of the 5,236 SNPs in our dataset, 2,461 of these were fixed for one island while being polymorphic for the other island. Of these 2,461 SNPs, 1,822 were fixed in the Kaikoura population, with the remainder being fixed in one, or both, of the North Island populations. This resulted in lower levels of observed heterozygosity in Kaikoura than in the North Island populations (Table 2, Fig. 3A). Inbreeding coefficient values were slightly lower in Kaikoura (Table 2, Fig. 3B). This is likely caused by greater habitat fragmentation (small rock platforms interspersed among sandy beaches) in the North Island populations than in the Kaikoura population (extended areas of intertidal rock platforms), which meant that isopods from the North Island sites were collected over smaller collection areas. Similarly, individual pairwise relatedness values were slightly lower for Kaikoura than for the North Island, although all values were generally low (Table 2). Mean pairwise relatedness between Hatfield’s Beach and Stanmore Bay individuals (r = 0.0174) was not significantly different to the mean relatedness within populations (r = 0.0171).

Table 2 Genetic diversity estimates showing the mean value with standard deviation (SD) for each population.

	Ho	He	FIS	r	
Stanmore Bay	0.223 SD 0.158	0.253 SD 0.163	0.118 SD 0.236	0.018 SD 0.01	
Hatfield’s Beach	0.229 SD 0.159	0.257 SD 0.162	0.107 SD 0.235	0.016 SD 0.01	
Kaikoura	0.153 SD 0.187	0.163 SD 0.190	0.059 SD 0.237	0.011 SD 0.02	
Notes.

Ho observed heterozygosity

He expected heterozygosity

FIS inbreeding coefficient

r TrioML pairwise genetic relatedness

Different approaches to resolving population structure were in close agreement: all analyses indicated very little genetic differentiation between the two North Island populations, but large and significant differentiation between the North Island and South Island. FST indices indicated strong population divergence both globally (FST = 0.31 p < 0.0001), and between the Hatfield’s Beach and Kaikoura (FST = 0.42 p < 0.0001), and Stanmore Bay and Kaikoura (FST = 0.42 p < 0.0001). In contrast, no genetic differentiation was identified between the North Island populations (FST < 0.0001, p = 1.0). A PCA of genetic distances revealed a clear genetic distinction between North Island and South Island populations, with the first principal component (PC) explaining 27.8% of the total variance and only 1.2% of the variance explained by the second PC (Fig. 4A). In contrast, a PCA for the two North Island populations did not further resolve population groupings, with PC1 and PC2 each explaining only 2.1% of the variance (Fig. 4B).

Hierarchical AMOVA analysis in Arlequin was concordant with the PCA and showed that 39.8% of the genetic variation occurred between the North and South Island groups and 59.9% within populations. In contrast, genetic variance between the North Island populations explained only 0.3% of the total variance. The genetic structure found between populations (FST = 0.40, p < 0.0001) and between groups (FCT = 0.6, p < 0.0001) was strong and directly comparable. No significant genetic structure was identified between the North Island populations (FSC = 0.004, p = 0.6). The amova function in the R package pegas produced similar results with the variance among groups accounting for 68.1% of the total variance (p < 0.0001), while the variance within groups accounted for only 0.19% of the total variance (p = 0.9).

Figure 3 Population genetic diversity statistics.

Tukey’s boxplots showing (A) mean observed heterozygosity and (B) the inbreeding coefficient for each population.

Figure 4 Principal coordinates analysis (PCA) showing individuals coloured by population.

(A) PCA with all populations showing distinct clustering of the Kaikoura population with the first and second principal coordinates explaining 27.8% and 1.2% of the variance respectively. (B) PCA for the North Island populations only with the first and second principal coordinates explaining 2.1% and 2.1% of the variance respectively.

Bayesian clustering analysis identified strong genetic structuring between the North and South Island populations with strong support for K = 2 (Fig. 5A). No apparent genetic structure was present between Hatfield’s Beach and Stanmore Bay. To be confident that strong hierarchical structure between the North and South Island was not obscuring any potential to identify weak structure in the North Island populations, we re-ran the analysis without the Kaikoura population using the LOCPRIOR model for weak genetic structuring. However, K = 2 was not supported by this analysis (Fig. 5B). Although ΔK suggested that the true K was four (Fig. S1), this statistic is unable to find the best K when true K = 1 (Evanno, Regnaut & Goudet, 2005), and inspection of the barplot of K = 4 did not reveal any genetic clustering (Fig. S2). Therefore, K = 1 for the North Island populations was inferred from the maximal mean log likelihood value for this K (Fig. S3).

Figure 5 Barplots of STRUCTURE results based on Bayesian clustering algorithms.

Barplots have been summarised over 10 replicates and show proportional membership of (A) 94 individuals of I. armatus to K = 2 genetic clusters and (B) 32 individuals from Stanmore Bay and 31 individuals from Hatfield’s Beach to K = 2 genetic clusters. Colours indicate proportional membership to a genetic cluster. Individuals are ordered by population of origin with vertical bars representing a single individual. NI, North Island, SI, South Island.

Patterns of gene flow between populations were inferred through model selection using Bayes factors in MIGRATE. This analysis revealed that the model representing unidirectional gene flow from the South Island into the North Island had the strongest support. The immigration rate (M) of South Island individuals into the North Island was 88.3 (95% Highest Posterior Density (HPD): 3.3, 166.7). When we considered only the two North Island populations, the full model with bidirectional gene flow between the two populations provided the best fit. Estimated migration rates were similar in both directions, with M = 1271.7 (95% HPD: 1176.7, 1360.0) from Stanmore Bay into Hatfields Beach, and M = 1205.0 (95% HPD: 1116.7, 1293.3) from Hatfields Beach into Stanmore Bay (see Figs. S4–S9 for posterior distributions of migration parameters for each model).

FST outlier tests

FST outlier tests of selection between morphotypes in BayeScan and Arlequin displayed very similar results. Within the analysis of the North Island populations only, the same six loci were identified as showing signatures of being under positive selection after controlling for the FDR in both BayeScan and Arlequin (Table S2). Two of these loci were highly significant in both analyses with q-values of zero. When all populations were considered, four loci were deemed under selection for morphotype after controlling for the FDR in both analyses (Table S2). This included the two highly significant loci from the North Island analysis. Visual inspection of the genotypes of these two loci revealed that the existence of at least one non-reference SNP allele at these loci was associated with the expression of the striped morph in all striped individuals from all populations (N = 13), apart from one individual with an uncommon striped pattern from Stanmore Bay (Fig. 1G). No individuals of other morphotypes possessed SNP alleles at these loci.

A PCA of genetic distances for the North Island populations using all loci and with individuals coloured by morphotype did not reveal any substantial clustering by morph (Fig. S10). However, PCA for the North Island populations using only the six SNPs deemed under selection revealed substantial clustering by morphotype (Fig. 6). This clustering was particularly apparent for the striped morph. The first and second principal components explained 41.1% and 23.9% of the total variance respectively. A PCA was not conducted with the four loci under putative selection identified when all three populations are included, because a minimum of five loci are needed for SNPRelate to run.

Figure 6 PCA of genetic distance using the six loci deemed under positive selection for colour with individuals coloured by morphotype.

Analysis is for the North Island (Hatfield’s Beach and Stanmore Bay) individuals only. Plot shows the first two principal components (PCs). PC1 explained 41.1% of the variance, while PC2 explained 23.9% of the variance. Points have been jittered (factor = 100) to improve readability.

Discussion

This study is one of the first population genomic studies on a marine organism in New Zealand. Using GBS, we were able to obtain a substantially large number (4,000–5,000) of high quality SNPs. Our results show evidence of highly contrasting levels of population connectivity and gene flow associated with different spatial scales in I. armatus. We also identified several loci that are under putative selection for colour. Our findings provide a strong case that reduced-representation genomic capabilities can efficiently resolve both neutral and putatively non-neutral genetic variation in species inhabiting the dynamic marine environment, and for which there is no reference genome.

Population connectivity and gene flow

In order to glean deep insight into the processes influencing genetic variation within and between populations, the geographical scale of population sampling regimes should equate to the scale at which dispersal occurs (Broquet & Petit, 2009). By using a large SNP dataset and sampling across two contrasting spatial scales, we successfully identified a scale below which population differentiation is not apparent, and a scale at which population differentiation is clearly defined. While we cannot completely rule out the possibility that our SNP dataset was unable resolve fine-scale population structuring (DeFaveri et al., 2013; Hess, Matala & Narum, 2011), we consider this scenario unlikely due to the large number of SNP markers used in our study (Gärke et al., 2012; Haasl & Payseur, 2011; Morin, Martien & Taylor, 2009).

Direct development is purported to limit isopod dispersal and drive strong genetic differentiation among populations inhabiting the same coastlines (Baratti, Filippelli & Messana, 2011; Carvalho & Piertney, 1997; Hurtado, Mateos & Santamaria, 2010; Lessios & Weinberg, 1994; Markow & Pfeiler, 2010; Sponer & Lessios, 2009; Xavier et al., 2011). Microgeographic genetic variation has been recorded in isopods, particularly in species exhibiting habitat fragmentation and limited vagility (Jolly, Rogers & Sheader, 2003; Piertney & Carvalho, 1994). For example, in Jaera albifrons, significant differentiation occurs across five metre distances due to the patchy distribution of suitable habitats (Piertney & Carvalho, 1994). In I. armatus we hypothesised that we would see low, but significant, levels of genetic variation between two North Island populations 8 km apart. Instead, we found no evidence of genetic structuring between these populations, with all analyses indicating panmixia. There was also no difference in mean pairwise genetic relatedness within or between the North Island populations, suggesting that there is no decline in dispersal rates with distance between the two sites. Although little is known of the dispersal abilities of marine isopods around the New Zealand coastline, McGaughran et al. (2006) found no evidence of population subdivision in three species of coastal stream-dwelling isopod, Austridotea spp, along the east coast of the South Island. Thus it appears that New Zealand isopods are capable of substantial levels of dispersal along coastlines, despite their sedentary reproductive histories. It is worth noting that both the estuarine Austridotea and marine Isocladus can, and often, swim. Thus it is probable that this mobility, in combination with displacement driven by surface storm events, facilitates dispersal along coastlines.

In strong contrast to the North Island, population differentiation between Kaikoura and the two North Island populations was extreme, indicating very little gene exchange between the North and South Islands. Migration rates (M) between the North and South Island were 14× smaller than those between the North Island populations. Estimates of M can be converted to the number of effective migrants per generation (Nm) using the equation χNm = Θi × Mi→j in which Θ = Neµ; where Ne = the effective population size, and µ= mutation rate per generation per locus). This equates to around 31 (95% Highest Posterior Density (HPD): 27, 35) migrants per generation in each direction between North Island populations, and 2.1 (95% HPD: 0.07, 4.13) migrants per generation between the North and South Island populations. These values are consistent with values of Nm estimated at two similar spatial scales in other direct developing marine invertebrates; the scleractinian Balanophyllia elegans (Hellberg, 1995), and Ototyphlonemertes spp (Andrade, Norenburg & Solferini, 2011).

However, estimates of gene flow produced in MIGRATE should be treated with caution for several reasons. First, care is needed in interpreting Nm because of the potential uncertainty in the effective population size indicated by Θ. Second, MIGRATE does not currently consider a model of zero migration. Given our other results, and that the range of the North–South HPD intervals almost overlap zero, this might be the most plausible scenario for the North and South Island. Similarly, it is important to note that the North Island analysis does not distinguish between the full model and panmixia, which could be a more appropriate model given our data. Third, because MIGRATE assumes that populations have been in equilibrium for ∼4N generations (Kingman, 1982), shared ancestry associated with recent population splits will lead to inflated estimates of gene flow (Kuhner, 2009). This is likely to be particularly relevant for the North Island migration estimates. Fourth, migration rates are assumed to be temporarily stable and are calculated across time to the most recent common ancestor. Therefore, estimated migration rates may reflect historical, rather than contemporary, gene flow (e.g., Marko & Hart, 2011).

Patterns of gene flow in the marine environment can be influenced by oceanographic features such as biogeographical boundaries and patterns of coastal circulation (Bilton, Paula & Bishop, 2002; Perrin, Wing & Roy, 2004; Ross et al., 2009; Wares, Gaines & Cunningham, 2001; White et al., 2010). It remains to be determined whether gene flow between Auckland and Kaikoura is interrupted by the presence of geophysical barriers, or whether populations follow a graduated model of dispersal, such as that exhibited by isolation-by-distance and stepping stone models. Nevertheless, the genetic subdivision present in I. armatus is congruent with an observed main break in the distribution of Sphaeromatidae species, with many North and South Island species displaying their geographical limits in the Kaikoura region (Hurley & Jansen, 1977). Furthermore, this break coincides with a defined boundary between the two main New Zealand marine biogeographic regions based on macroalgal and macroinvertebrate taxa (Shears et al., 2008). This evidence suggests that there may be a geophysical barrier present near Kaikoura which poses a barrier to gene flow between North and South Island marine communities.

This north–south phylogenetic subdivision is a common finding emanating from marine population genetics studies in New Zealand (Apte & Gardner, 2002; Ayers & Waters, 2005; Goldstien, Schiel & Gemmell, 2006; Hickey et al., 2009; Keeney, Szymaniak & Poulin, 2013; Ross et al., 2012; Sponer & Roy, 2002; Veale & Lavery, 2011; Waters & Roy, 2004), particularly down the east coast of New Zealand where currents are stronger than on the west (Heath, 1985; Stanton, 1976). Previous work in direct developing marine invertebrates suggests that ocean currents and upwelling regions (Fig. 2) associated with the north–eastern coast of the South Island (Sponer & Roy, 2002; Keeney, Szymaniak & Poulin, 2013), or East Cape (Knox, Hogg & Pilditch, 2011; Stevens & Hogg, 2004; Veale & Lavery, 2012) could be responsible for these sharp genetic disjunctions. Although our current data do not allow us to identify the location of the genetic break, our study suggests that a strong barrier to gene flow is present between Auckland and Kaikoura, and advocates for future intensive sampling between these two sites. If these intermediate sites are combined with a hierarchical sampling regime that includes populations sampled across staggered spatial scales, I. armatus’ capacity for dispersal, as well as the influence of any hydrographic features on gene flow, can be determined.

Similar to findings in other intertidal and deep-sea isopods (e.g., Baratti, Filippelli & Messana, 2011; Raupach et al., 2007; Sponer & Lessios, 2009; Xavier et al., 2011), and as suggested for some NZ marine invertebrates exhibiting similar North-South subdivision (e.g., Sponer & Roy, 2002; Stevens & Hogg, 2004), the strong genetic divergence between the North and South Island populations alludes to the possible existence of cryptic species. Sanger sequencing of the cytochrome c oxidase subunit 1 mitochondrial gene should be conducted to assess this possibility (Lefébure et al., 2006).

If I. armatus does indeed represent a cryptic species complex, the evolutionary persistence of morphotypes across speciation events suggests that polychromatism in I. armatus is likely maintained by selection. The alternative-that shared polychromatism is due to demographic processes such as the retention of ancestral polymorphisms or introgression-is unlikely since we found no evidence of genetic admixture within North and South Island individuals, and given the large number of SNPs that were fixed within North and South Island populations. However, further refinement of rates of gene flow and analysis of divergence times between the two main islands would help distinguish between these processes.

Genetic basis of colouration

We found possible evidence of a genetic basis to one component of CP. Six SNPs were deemed to be associated with, or are in linkage disequilibrium with, a gene that is under putative positive selection for morphotype. Of these six SNPs, two showed evidence of an association to the expression of the striped morph. Our findings suggest that there may be more than one locus controlling the phenotypic expression of the white stripe because one striped morph with an uncommon patterning and less-defined stripe did not possess the SNP allele at these loci. Although the diverse colouration in I. armatus suggests that the genetic basis of colouration in this species is likely to be complex, our findings suggest that at least some polymorphisms may be controlled by a relatively small number of loci (e.g., Sassaman & Garthwaite, 1980), and that their effects on phenotype could be additive. In order to confirm a genetic basis to polychromatism in I. armatus, our analyses require verification with larger sample sizes for each morph, accompanied by captive breeding experiments based on Mendelian genetics. Presently, the adaptive function of colouration in I. armatus is unknown. In isopods, however, predation is a common driver of stabilising selection on colouration (e.g., Jormalainen, Merilaita & Tuomi, 1995; Merilaita, 2001), which should fix colouration onto the single most efficiently cryptic phenotype for that population. We know that small-scale variations in colour morph frequencies can occur in I. armatus (Jansen, 1971). Therefore, the high gene flow occurring at relatively small spatial scales (<8 km) demonstrated by our study suggests the existence of selective pressures that can maintain CP within populations in this species. An integrated knowledge of population connectivity with a detailed analysis of the naturally occurring selection associated with phenotypic variability is needed to identify the selective pressures acting on colouration. To achieve this, gene flow will need to be resolved over more populations and compared with an analysis of colour variation at each site.

Conclusions

Our study provides evidence that GBS can efficiently resolve both fine-scale and broad-scale genetic structuring in a non-model vagile marine species. Our findings suggest that I. armatus possesses substantial dispersal ability despite lacking a pelagic larval stage. However, in accordance with studies in other New Zealand marine invertebrates, strong differentiation between North and South Island populations suggests the presence of at least one barrier to gene flow down the east coast of New Zealand. This study improves our understanding of the fine-scale dispersal capabilities of marine invertebrates lacking a pelagic larval phase, and provides a platform for further investigations into the role that gene flow plays in maintaining CP in this species.

Supplemental Information

Table S1 Quality filtering applied to the 26,215 putative SNPs identified by GBS

Click here for additional data file.

Table S2 Loci deemed under selection for morphotype after FDR correction, for FST outlier analyses conducted in BayeScan and Arlequin

Analyses were run with all three populations, and with only the two North Island (Hatfield’s Beach and Stanmore Bay) populations. Loci are listed in order of decreasing significance.

Click here for additional data file.

Figure S1 Plot of ΔK values for each K (number of clusters) output from STRUCTURE HARVESTER from a STRUCTURE run of the two North Island populations

Click here for additional data file.

Figure S2 Barplot showing proportional membership of North Island individuals to four genetic clusters, as identified by ΔK values.

Colours indicate proportional membership to a genetic cluster summarised over 10 replicates in the program CLUMPP. Individuals are ordered by population of origin with vertical bars representing a single individual.

Click here for additional data file.

Figure S3 Scatterplot showing mean log likelihood values with standard deviation (SD) for each K output from STRUCTURE HARVESTER for the STRUCTURE analysis of the North Island populations

Click here for additional data file.

Figure S4 MIGRATE posterior distribution from the unidirectional model (M1→2) for the North Island analysis showing migration estimates (M) from population 1 (Stanmore Bay) into population 2 (Hatfield’s Beach)

Grey shaded area shows the 95% HPD intervals. Dashed line represents the mode of the posterior.

Click here for additional data file.

Figure S5 MIGRATE posterior distribution from the unidirectional model (M2→1) for the North Island analysis showing migration (M) estimates from population 2 (Hatfield’s Beach) into population 1 (Stanmore Bay)

Grey shaded area shows the 95% HPD intervals. Dashed line represents the mode of the posterior.

Click here for additional data file.

Figure S6 MIGRATE posterior distributions for the full model with bidirectional migration (M) for the North Island analysis showing migration estimates from (A) population 2 (Hatfield’s Beach) into population 1 (Stanmore Bay), and (B) population 1 into popula

Grey shaded area shows the 95% HPD intervals. Dashed line represents the mode of the posterior.

Click here for additional data file.

Figure S7 MIGRATE posterior distribution from the unidirectional model (M1→2) for the North-South Island analysis showing migration (M) estimates from population 1 (North Island) into population 2 (South Island)

Grey shaded area shows the 95% HPD intervals. Dashed line represents the mode of the posterior.

Click here for additional data file.

Figure S8 MIGRATE posterior distribution from the unidirectional model (M2→1) for the North-South Island analysis showing migration (M) estimates from population 2 (South Island) into population 1 (North Island)

Grey shaded area shows the 95% HPD intervals. Dashed line represents the mode of the posterior.

Click here for additional data file.

Fugure S9 MIGRATE posterior distributions from the full model for the North-South Island analysis with bidirectional migration (M) showing migration from (A) population 2 (South Island) into population 1 (North Island), and (B) population 1 into population 2

Grey shaded area shows the 95% HPD intervals. Dashed line represents the mode of the posterior.

Click here for additional data file.

Figure S10 Principal coordinates analysis showing the first and second principal coordinates for the North Island populations with individuals coloured by morphotype

The first and second principal coordinates explain 2.1% and 2.1% of the variance respectively.

Click here for additional data file.

We would like to acknowledge Landcare Research Ltd. and the University of Canterbury for providing laboratory and field logistical support respectively. We are grateful to David Eme for providing helpful suggestions and some of the scripts used in the analyses; and Libby Liggins and David Aguirre for constructive advice on this project. Thanks also to Julia Allwood, Ana Ramón-Laca, and Duckchul Park for laboratory advice and support. We thank Shane Lavery and an anonymous reviewer for their invaluable reviews that improved this paper, and Peter Prentis for his constructive comments as editor.

Additional Information and Declarations

Competing Interests

Author Contributions

Field Study Permissions

Data Availability

The authors declare there are no competing interests.

Sarah J. Wells conceived and designed the experiments, performed the experiments, analyzed the data, prepared figures and/or tables, authored or reviewed drafts of the paper, approved the final draft.

James Dale conceived and designed the experiments, approved the final draft, edited the manuscript.

The following information was supplied relating to field study approvals (i.e., approving body and any reference numbers):

Field experiments were approved by the Ministry of Primary Industries, New Zealand (Special Permit 632).

The following information was supplied regarding data availability:

NCBI Short Read Archive: Bioproject PRJNA486197. The metadata and raw sequence reads have also been uploaded to GeOMe (https://geome-db.org/query) under the expedition name Isocladus_armatus_DArTseq.

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
