# Peer review of "Contrasting gene flow at different spatial scales revealed by genotyping-by-sequencing in Isocladus armatus, a massively colour polymorphic New Zealand marine isopod"

_PeerJ, doi:10.7717/peerj.5462_

## Round 0.1 · original submission · Major Revisions

Overall, I found this an interesting manuscript on an interesting topic in population genomics. Currently, as it stands, this manuscript needs revision before it can be acceptable for publication. In particular three areas need to be revised in some detail. The areas for revision are:
1. I agree with reviewer 1 that the colour polymorphism seems a little oversold in this preliminary manuscript. Currently, I think your ability to determine what evolutionary factors are maintaining this polymorphism is a bit weak. I do feel that you can make preliminary inferences about alleles found only in certain CPs and that may be contributing to CP in this species. I suggest rewording the introduction to reflect this difference.
2. I also agree with reviewer 2 that your conclusions about the north south division need more intensive sampling of populations. This should be reduced in the discussion and highlight the spatial scale at which population structure occurs more.
3. Finally, the program Migrate can be a little unreliable at estimating gene flow because of the assumptions indicated by reviewer 1. I think if you use this analysis you need to include some of the caveats in the discussion section.

Please reply to all comments of both reviewers (who did a great job) in your revised manuscript.

Reviewer 1 ·

Basic reporting

Refer to General Comments

Experimental design

Refer to General Comments

Validity of the findings

Refer to General Comments

Additional comments

The authors use a reduced representation genomic approach to investigate the hypothesis that colour polymorphism within populations of the isopod I. armatus is maintained by the action of divergent selection coupled with gene flow between populations. The authors are successful in validating the use of GBS markers to infer populations structure within and between Southern and Northern Island populations in New Zealand and they use these data to compare coalescent models differing in the direction of migration across regional and coastal spatial scales. The authors also demonstrate that few outlier loci identified as putatively under selection appear to be associated with distinct colour morphs. The mechanisms maintaining color polymorphisms in populations are certainly important evolutionary questions and I. armatus appears to be a useful model for testing such hypotheses. While individual genomic analyses seem to be well done and appropriate, I am not convinced that the hypothesis of interest can be explicitly tested from the data and analyses provided. There are several factors that weaken the study and the conclusions that can be drawn about the nature of selection and the impact of gene flow on CP. Nevertheless, the application genome-wide markers to infer population structure and make inferences about dispersal ecology in this system are useful and interesting and will be a valuable contribution if appropriate revisions are made.

Major Comments:

My main concern is that this study is framed in the context of understanding the evolutionary mechanisms maintaining colour polymorphism; however, the study design does not set up a test of this hypothesis, which would require both variation in the frequencies of colour morphs among populations and sampling populations with and without gene flow. No justification is provided on why selection is expected to act differently among populations in this system and it is not evident whether distinct colour morphs segregate between populations or habitats within populations. Instead, the discussion of the empirical evidence suggesting that selection may be operating on CP in I. armatus seems post hoc (L457). Furthermore, the effect of gene flow between populations and the processes occurring within populations that may contribute to the maintenance of CP cannot be detangled with the data at hand. More broadly, the mechanism by which directional selection and gene flow may act together to maintain CP within and between population is not clearly explained in the Introduction. Explicit predictions of this hypothesis would be useful.

Overall, while the results generally support the conclusions made, they do not match the early focus on CP and the processes maintaining CP. This mismatch is further evident in the Discussion, where emphasis is placed on the broader context of interpreting population structure and barriers to dispersal, rather than the implications for the evolution of CP. With that being said, I would suggest that the aims of the study relating to CP need to be removed entirely or revised to explicitly make this aim secondary to the main investigation.

My second concern is that MIGRATE model comparisons of gene flow do not allow for m=0 as the authors indicate. Plots of the posterior distributions of the migration parameters estimated by MIGRATE (prior to model selection) should be provided in the supplementary materials at the very least. It seems essential that specified priors capture the posterior distributions for accurate estimates of migration and therefore model support (L274). I would also suggest to draw direct comparisons of estimated m (the proportion of migrants per generation) between population pairs, rather than Nm, to avoid uncertainty in effective population size estimates.

The authors should also keep in mind that MIGRATE involves the assumption of migration-drift equilibrium such that all shared alleles between two populations are due to gene flow (refer to Kuhner 2009 TREE 24.2:86-93). Because your data includes populations distributed across small spatial scales that may have split only very recently (<4Ne generations), it is likely that some variants are shared by descent, rather than migration. Violations of these assumptions may therefore inflate estimates of gene flow, so it would be useful to touch on these caveats in your discussion. Consideration of demographic processes leading to population structure (i.e., historical isolation, rather than contemporary barriers to gene flow) should be given and these caveats should be discussed.

Additional Comments:

L40-41. Statement seems too general, as there are many examples of marine species with low capacities for dispersal.

L43-55. Discussion and references on the application of genomics in marine systems seem dated.

Figure 1. It would be useful to arrange images to allow direct comparison of colour morphs (i.e., striped morphs next to each other). Colour morph of (h) is not indicated in figure legend.

L82-85: Given the apparent continuous distribution of colour variation within populations, CP is likely to have a polygenic basis, and such caveats should be addressed.

L164: Unclear whether ‘call rate’ refers to mapping quality or ‘missingness’ across individuals.

L204: Some justification for including COANCESTRY analyses would be useful.

L227: Unclear which SNP dataset was used in the PCA analysis, and whether missing genotypes were removed following SNP filtering.

L356-358: Unclear how the strength of associations of two SNPs with striped colour morph was determined. Small sample sizes of the striped morph in particular may inflate false positive associations.

Figure 6 is missing. Based on the text alone it is unclear whether the PCA of selected SNPs was conducted for colour morphs across all populations and if the clustering of morphs is consistent when all samples from populations are considered.

L368: Unclear why emphasis on small marine species is relevant here.

Table 5. It seems unusual that more than 70% of the SNPs analysed in neutrality tests (3,892/5,236) are FST outliers among populations. Is this the total number of SNPs under selection in both FDIST and BAYESCAN analyses combined?

Table S1: Unclear what the values refer to. Locus ID? Table should include a legend.

L450-452: The interpretation that CP is maintained among species through adaptive processes entirely, seems speculative. Justification is required to explain why the maintenance of CP among species solely through demographic processes does not apply in this system - namely ancestral polymorphism, drift, migration in the absence of divergent selection.

·

Basic reporting

This is a very clear and succinct paper addressing a preliminary study of population structure in a New Zealand marine isopod using population genomic techniques. It is an interesting topic, especially given the extreme colour polymorphism in the species, the methodological approach is appropriate, the writing is very clear and logical, and most of the conclusions are appropriate and justified.

The abstract is clear and succinct. The Introduction covers all the basic required background elements of marine population genetics, isopods, population genomics, and the genetics of colouration, and introduces the aims in a succinct manner. The Results are sufficiently presented and illustrated through tables and figures. The Discussion is appropriate and justified, apart from the points mentioned above. The referencing is quite comprehensive for a preliminary study.

Experimental design

The methods all appear appropriate and are well described.

Validity of the findings

My only criticism of this manuscript is that a reasonably large section of the discussion is spent on potential support in this data for a north-south genetic division, which has been seen in some other marine species in this region. My take on it is that, given the preliminary sampling analysed here, although the north-south division is relevant, this data set cannot yet really add anything to this debate. Thus this discussion should be reduced to a recognition of this existing pattern, and how it could best be tested in the future in this species. Instead, I think the authors should emphasise the obvious population strength of this preliminary data set, which is that their sampling at two different geographic scales has enabled them to already determine two important aspects of this species’ population structure: - a scale below which population differentiation appears to not occur (8km), and a scale at which population structure clearly does occur (North v South I.). This is exactly what is required from a preliminary study of this nature.

The authors also appropriately make two other preliminary conclusions from the data. That is, that the methodological approach provides appropriate data on genetic diversity for undertaking population structure analyses, and that, significantly, there is already some preliminary evidence that some of the colour polymorphism may be linked to simple genetic variation.

The claim that this is the first population genomic study on a NZ marine organism may not be justified (e.g., Villacorta-Rath et al, 2016 – see below), so I think it would be more appropriate to simply say it is one of the first studies.


Some detailed comments follow:


Line 26: - “ …one of the first …”

37: - … remove most variation…

56: - not the first genomic study (see e.g., Villacorta-Rath et al 2016 below) – perhaps the first reported GBS study (depending on the definition of GBS). Probably safer to simply say “…one of …”
[Villacorta-Rath C, Ilyushkina I, Strugnell JM, et al. (2016) Outlier SNPs enable food traceability of the southern rock lobster, Jasus edwardsii. Marine Biology 163.]

69: - “…NZ marine Isopoda…”


Fig 2:
A – should be a table? Or actually incorporated into the fig.?
B – the choice of colour of landmass (light blue) may perhaps not be the best contrast for publishing

150: - “…were processed…”
155: - “…a cheaper version…”
176-200: - a pop structure analysis on those loci potentially under selection may also be interesting to see at a later stage

Table 1 could go into appendices?

Table S1 requires additional explanatory text as fig legend (in my copy)

I don’t see fig 6 – it should be either included or reference to it removed

366: - I don’t think this is true – see above for 1 e.g.,

**397-441: - although interesting, and a good reflection of the current marine phylogeographic literature in NZ, given the sampling, a lot of this discussion is a bit speculative & somewhat irrelevant– can only say that sig. differentiation occurs at a scale >8km

Better to focus on the strength of analysis of different scales, & potential correlation of some markers with morphology

Good conclusions

Additional comments

This is an excellent preliminary paper and I look forward to seeing progress on this study.

---

## Round 0.2 · accepted · Accept

Overall, this is an interesting and well thought out paper that I feel will make a good contribution to PeerJ. I agree with the reviewer that the manuscript is fine to be accepted, but needs a few small areas tidied up. Please see their review for the few points to correct while in production.

# Reviewer 1 ·

Basic reporting

Refer to general comments to the author.

Experimental design

Refer to general comments to the author.

Validity of the findings

Refer to general comments to the author.

Additional comments

I have re-read the revised manuscript carefully and I feel that the authors have addressed my comments and the suggestions from the other reviewer appropriately. I would recommend the revised paper for publication.

Overall, the revised Introduction and study aims (i.e., shifting the primary focus away from CP) are much better suited to the results and the conclusions of this study. The authors’ scaled back interpretation of the results relating to selection and the limited inferences that can be made about the maintenance of CP seem more appropriate. The assumptions and caveats regarding Migrate analyses are well written. I also appreciate the additions of supplementary figures illustrating the posterior distributions of the estimated migration parameters. I have a few suggestions for minor clarifications below.

183-184: Unclear filtering criteria regarding linked loci. Clarification would be useful.

276: What is the generation time assumed for this species?

387: putatively non-neutral

407: More explicit statement of this hypothesis would be useful in the Introduction. It is unclear whether you predicted to observe genetic structure at small spatial scales in this species as you argue for both strong swimming ability and previous studies indicating substructure across microhabitats.

474: Seems unlikely, given that such patterns are reported among northern and southern populations of other species in this region. I agree that sequencing COI will be useful prior to future analyses of CP in this system.

488: one component of CP